# Intra-Rater Test-Retest Reliability of a Modified Child Functioning Module, Self-Report Version

**DOI:** 10.3390/ijerph17196958

**Published:** 2020-09-23

**Authors:** Kwok Ng, Piritta Asunta, Niko Leppä, Pauli Rintala

**Affiliations:** 1School of Educational Sciences and Psychology, University of Eastern Finland, 80101 Joensuu, Finland; 2Physical Activity for Health Research Cluster, Department of Physical Education and Sport Sciences, University of Limerick, Limerick V94 T9PX, Ireland; 3LIKES Physical Activity for Health Research Centre, 40700 Jyväskylä, Finland; piritta.asunta@likes.fi; 4Spesia Vocational College, Keskussairaalantie 21, 40620 Jyväskylä, Finland; niko.leppa@spesia.fi; 5Faculty of Sport and Health Sciences, University of Jyväskylä, 40014 Jyväskylä, Finland; pauli.rintala@jyu.fi

**Keywords:** disability statistics, kappa, intraclass correlation coefficient, young adolescents, functional difficulties, special education, survey, health behavior

## Abstract

Determining disability prevalence is a growing area for population statistics, especially among young adolescents. The Washington Group on Disability Statistics is one source of reporting disabilities through functional difficulties. Yet, young adolescents self-reporting through this measure is in its infancy. The purpose of this study was to carry out an intra-rater test-retest reliability study on a modified set of items for self-reporting functional difficulties. Young adolescents (*N* = 74; boys = 64%; age M = 13.7, SD = 1.8) with special educational needs in Finland completed a self-reported version of the Child Functioning Module in a supervised classroom. The second administration took place two weeks later. Intraclass correlation coefficient (ICC) and Kappa (*k*) statistics were used to test the reliability of the items, and interpretation took place through Landis and Koch, and Cohen, respectively. The majority of items had substantial or moderate agreement, although there was only fair agreement for self-care (ICC = 0.59), concentration (ICC = 0.50), and routine changes (ICC = 0.54). Kappa statistics of behavior control were interpreted to be large (*k* = 0.65), and seeing (*k* = 0.49), walking (*k* = 0.49), and speaking (*k* = 0.49) difficulties were moderate. The majority of the items in the self-reported version of the Child Functioning Module can be used in a scale format, although some caution may be required on items of self-care and concentration when used as a dichotomous variable.

## 1. Introduction

Based on the Salamanca Agreement on Inclusive Education, all children have the right to education, irrespective of individual difficulties [1]. Since then, the Finnish education system has been progressing towards more inclusion in schools by passing the Education Act in 2010, where families have the choice for children to attend a general school, special educational class, or special school [2]. Changes to the educational structures has seen a year-on-year increase in the number of children in general schools who require special or intensified support from 8% in 2010 to 20% in 2019 [3].

A multi-tiered framework explains this big rise. In Finland there is a three-tier support system, with the purpose of support learning at the earliest possible opportunity for the child and to be within inclusive environments. The Basic Education Act [2] and the three-tier framework was officially implemented in August 2011 in every Finnish school [4]. The support system allows these pupils to become part of the general school, be in environments whereby they have access, and can participate in the same activities as their peers. This type of support is described as Tier 1—general support. In Tier 1, the support level is offered for every pupil in the Finnish education system, Tier 2 is intensified support, and in Tier 3, pupils are given special support.

In addition to monitoring academic progress, schools are a good place to recruit children for important health checks as well as carry out health surveys. Monitoring tools of health behaviors should also include children with support needs [5]. However, few instruments do this. The majority of surveys often exclude children with disabilities [6], which may lead to response bias and a knowledge gap when it comes to national reporting. Furthermore, completion of survey instruments may be inappropriate for children with support needs, and thus a gap in knowledge of health behaviors among children with disabilities exists.

### Difficulties in Measuring Disabilities

Conceptually, the measurement of disabilities has its difficulties [7]. There is often a stigma related to the reporting of disabilities and short measures often lack detail to understand which features make a person feel like they have a disability [8]. To address these previously reported issues, items based on The World Health Organization (WHO) International Classification of Functioning, Disability and Health (ICF) are used as indicators for disabilities [9,10]. Core functions that influence children’s development, based on the Washington Group on Disability Statistics short set [11], have been created with the assistance of United Nations International Children’s Emergency Fund (UNICEF) [12], to present the Child Functioning Module (CFM) [13]. Additional functions that are crucial in the development of children under the age of 18 years that are distinctly different from adulthood, include psychosocial development items in learning, behavioral control as well as social interactions that were added in the CFM [14]. Although there has been a number of studies that have tested the viability of the CFM [15,16,17], these were primarily based on the proxy version of the questionnaire set (i.e., completed by parents). It is not known if these instruments can be used in the context of children taking part in a self-reported survey.

There is the burden on adults to complete surveys on behalf of children and the lack of self-reporting from children is a violation of their basic human rights [18]. Adolescents need to be able to self-report their own overall health (physical, mental and social) with such information often referred to as health-related quality of life [19]. It is not uncommon for adolescents with disabilities to report lower ratings of their own health-related quality of life [20]. Self-reporting of health-related quality of life is a predictor of temporal functioning; however, details of specific fixed impairments are often neglected in research [19]. Therefore, it is essential that other health-related data are collected. For health behavior surveys, it is imperative to have reliable instruments as part of the validation process. Intra-rater reliability can be carried out through a test-retest mode, whereby participants carry out the test twice [21]. Completion of the test-retest can yield recency effects, whereby responses reflect on memory of responses rather than on reporting on actual behaviors [22]. However, too much time between survey completion may generate true changes in the responses due to behavioral changes and that would alter the test-retest scores [23]. Given the importance of accurately measuring disabilities among children with special support needs, and the lack of psychometric properties available from the self-report version of disabilities from surveys, the aim of this study was to carry out a test and retest reliability study on the self-report version of the CFM among children with special educational needs in schools.

## 2. Materials and Methods

The study had received approval by the University of Jyväskylä ethics committee. According to the Finnish Ministry of Education school lists, there are 60 schools with a special education status. The location of the schools was examined, and a convenience sample was selected based on the schools clustered in one region of Finland. A one-tail test with power at 0.80, alpha at 0.05 and 0.30 as the hypothesized level of correlation, specified that the target sample size needs to be 67 students [24].

### 2.1. Procedures

Schools in the allocated region (*n* = 10) were contacted. A researcher (NL) described the procedures of the study and asked if it was possible to obtain permission to take part in the study. Schools who agreed (*n* = 4) to take part in the study received equipment for adapted physical education and sports as a token of gratitude. School principals selected a class in the school with children in equivalent grades of the ages 11 years, 13 years, and 15 years old. This age range was chosen as other items in the questionnaire were appropriate for young adolescents in other national and international health behaviors in school-aged children surveys conducted by the authors [25,26]. Principals were asked to make a list of pupils who would be able to complete a survey independently (who had the ability to read questions and enter responses on a computer by clicking a mouse) and then randomly selected the pupils.

Researchers visited the school site to administer the online surveys that consisted of various questions on adolescent health and physical activity behaviors. The class teachers were given a short website address link to give to each of the pupils. There were different links depending on the age or ability of the pupils. The teachers encouraged the pupils in a class to take part in the study but were not forced to complete it and it was carried out anonymously. Researchers were present to give instructions to the pupils, teachers, and teacher assistants before telling the pupils they could start the survey. Teachers were asked to complete their own version of the survey, allowing the perception of confidentiality during completion of surveys. Some of the children had personal assistants with them, and some other children shared an assistant. Pupils entered their responses on the computers by themselves. Students were permitted to ask teachers, assistants, and researchers to clarify some items they did not understand, at times the assistants may have read aloud the question directly to pupils. Teachers and researchers were instructed from the protocol to avoid answering it for them. Some pupils needed specific clarification for abstract questions, for example, Cantril’s life satisfaction ladder [27]. These items were not included in this intra-rater test-retest study, only the CFM.

Teachers were asked to allocate a choice of four surveys to the pupils based on the age and developmental stages of the individual. The surveys were; (1) Long survey (L) with 60 questions targeted at pupils aged the equivalent of 15 years; (2) an easy-to-read modification of the long (L-er) survey targeted at pupils aged the equivalent of 15 years but with basic language requirements; (3) a medium survey (M) with 40 questions targeted at pupils aged the equivalent of 11–13 years; and (4) an easy-to-read modification of the medium (M-er) survey targeted at pupils aged the equivalent of 11–13 years but with basic language requirements. The reduction of items between the two age groups was based on the experiences of the survey design from the WHO Collaborative Health Behavior in School-aged Children (HBSC) study [28].

The L and M versions of the survey were sent to the Finnish Easy-to-Read service to make the changes to the question items. The items were then sent back to the research team for consideration. Modifications continued until there was agreement between the Easy-to-Read service and the researchers so there would be consistency with original and modified constructs. Although there were differences in the number of questions in M and L, the placement of the CFM was the same, with both at the beginning of the surveys. Placement at the beginning of the survey reduced the way completion of versions of the survey affected the aims of the study.

The pupils completed the survey independently on two occasions. The time between surveys was two weeks. Surveys were completed through an online survey platform. However, for part of the first data collection date, there were server outages and for those participants (*n* = 14), the survey was carried out by pen and paper (print out of the online survey) and coded in by the researchers. Subsequent surveys were completed through the online survey. There were further server outages during the data collection period; however, responses were refreshed in order for the data to be saved. 

### 2.2. Measures in This Study

Pupils entered their sex (boy or girl), their month, and their year of birth. A calculation was made based on the time of survey completion to create an age variable.

The CFM was derived from the joint work of the Washington Group on Disability statistics and UNICEF [14]. However, the original was modified in several ways that allow for cultural differences. The first modification was to transfer the content from proxy reporting (by parents) to self-report. For example, the original question would begin with “Does your child have difficulties in…” and the modified version became, “Do you have difficulties in…” The next modification was based on item reduction. The CFM has a layered approach to functioning, and the modified version was based on a single item per function. For example, the CFM has three items related to the seeing function. The first item is a screener for whether the child uses glasses or contact lenses, and then depending on the answer, there is a skip function to assess the difficulty in seeing. The modified version we used was a single item about “seeing difficulties, even if the child wears glasses or contact lenses.” This type of modification has been used in the development of the Washington Group Short Set questions [11]. The third modification was to group the items together to give the impression that the child was answering fewer questions. In the CFM, there are separate questions for each function. In the modification, the same header was used, “Compared to children of the same age, do you have difficulties in…”, and then the corresponding functions were listed. This was the presentation of the items in the L and M version. The entire sentence was included in the easy-to-read versions. The differences between the L and M versions and the L-er and M-er versions are shown in the Table 1.

All items had a four-category response scale with the following options, “None”, “Some”, “A lot”, and “Cannot do”. Translations of the items were carried out with contextual back translations for items that were not already available in Finnish and reported elsewhere [29]. The development of translation of the original items closely followed the instructions from the Washington Group for translations. Unlike direct back translations, contextual translations take into count the context of the local language during the translation process [30]. The translations were corrected until experts in disability and adolescence surveys (KN, PR, NL, PA) were satisfied that the items in Finnish matched the original items. Moreover, in the translation process, visual representations of the response scales were used to help the respondents to understand the differences between the response options. They were color-coded from green for “None”, orange for “some”, red for “a lot”, and a cross for “cannot do”. 

One final modification was made to this self-report version of the CFM. The CFM has items related to mental functions [9]. One item is related to the functions on being very anxious, nervous or worried, and the other item is related to being sad or depressed. The response scale in the CFM is different to the other functions, whereby questions were related to frequency of mental dysfunction. This is because corresponding responses for items on mental dysfunction would be difficult to comprehend, whereas the frequency of recalling symptoms is a reliable method among populations who complete the survey [31]. Based on earlier research on a psychosomatic symptom checklist, the two items closest in relation to these two items were also included in the survey [32]. The items used were headed with the following, “How often have you had the following symptoms over the past 6 months? Tick one box for each symptom”. The symptoms listed were depression or feeling low and nervousness. The response scale included the following: “almost daily”, “more than once a week”, “approximately once a week”, “approximately once a month”, and “less or never”. Due to the differences in the way the CFM was used in our study, these results were not reported.

### 2.3. Analyses

The survey data combined the test and retest surveys. A unique identifier was coded for each participant in each survey. Time for completion was not analyzed as this was not available due to there being other questions in the overall survey. Data from participants who completed both surveys were included in the final data sheet. The data were imported into IBM SPSS version 24.0. (IBM Corp, Armonk, NY, USA) for statistical analyses. Reliability between test and retest was computed through the single measure of intraclass correlation coefficients (ICC). The two-way random model with absolute agreement type was performed, and test statistics were set to 95% confidence intervals (CI). Acceptable reliability criteria were based on the Landis and Koch divisions of agreement [33]. To interpret the categories, the following were used: less than 0.20, slight or poor; 0.21–0.40, fair; 0.41–0.60, moderate; 0.61–0.80, substantial; and over 0.80, almost perfect.

Single functions were also dichotomized to test various cut points between a state of “disability” versus “no disability”. Two sets of cut-off values for each function were set to (1) at least “some”, and (2), at least “A lot” as guided by previous research [11]. To test this, the Cohen’s Kappa statistics were used to estimate the stability of each function. Cohen’s Kappa can be interpreted with the following correlation values, greater than 0.5 being large; 0.3–0.5 moderate; 0.1–0.3 small; and less than 0.1 trivial [34].

## 3. Results

### 3.1. Descriptive Results

The majority of the participants (*N* = 74) completed the M-version of the survey (Table 2). According to the cut-off points of at least some difficulties, to indicate disabling functions, almost two thirds of the respondents would be considered to have disabilities.

The prevalence of disability in the study varies depending on which cut-point is used (Table 3). The most common functional limitations where the individual has some difficulties were in the cognitive domain, such as difficulties in learning (32.4%) or in remembering (31.1%). The most common function adolescents reported they could not do (most severe limitation), was the domain of getting friends (5%).

### 3.2. Test-Retest Results

According to the interpretation by Landis and Koch [33], 6 of the 11 functions had substantial agreement after a two-week gap between completing the survey (Table 4). Difficulties in learning and difficulties in getting friends had moderate agreement. Three items (self-care, concentration, and routines changes) had fair agreement.

Kappa was tested on two cut-off points, at least some difficulties (Kappa 1), and at least a lot of difficulties (Kappa 2). According to the interpretation by Cohen and Cohen [21], five out of 11 functions had large (seeing, walking, remembering, behavior, friends) Kappa 1 values. There were four moderate (hearing, speaking, learn, routine changes) and two small (self-care, concentration) Kappa 1 values. There was one large Kappa 2 value for difficulties in behavior control. In addition, three other difficulties had moderate (seeing, walking, speaking) Kappa 2, three with small (learn, remembering, getting friends) Kappa 2, and three with poor (self-care, concentration, routine changes) Kappa 2 values.

Difficulties in seeing, walking, remembering and behavior control were performed consistently as an entire scale and as cut-off points used to determine disability classification. There was not enough test-retest data for pupils who reported difficulties in hearing to determine how well the test-retest performed. In other words, none of the individuals who reported at least a lot of difficulty in hearing during the test survey completed the retest survey.

Difficulties in remembering and getting friends functions had inconsistent results. There was substantial agreement across the scale of remembering difficulties, a large agreement for Kappa 1 values, but small agreement for Kappa 2 values. Difficulties in getting friends had large Kappa 1 values, small Kappa 2 values, and moderate agreement across the scale. Other subtle differences across the results were noted. Difficulties in self-care and concentration are items that have fair agreement and small Kappa 1 values. Difficulties in making changes to routines also had fair agreement, but the Kappa 1 value was moderate.

## 4. Discussion

To the authors’ knowledge, this is the first time the CFM has been tested without proxy in a special educational setting. The use of the CFM for self-reported disabilities is, overall, an acceptable measure. This is an important finding because previous work on these items has been based on proxy reporting [13,35] and there is a need to include self-reported disabilities in national health surveys [5]. Young adolescence is an important time for reflecting on personal growth and these changes must be monitored [19]. More specifically, in this study the items on the self-reported version of the child functioning module were completed by over 85% of the pupils with special support needs. There were specific items, most notably the item on “self-care”, “concentration”, and “routines changes” that may need to undergo further development to ensure acceptable reliability, especially in a Finnish special education setting. These new findings are discussed in this paper.

### 4.1. Reliability as a Scale

The self-reported version of the CFM was designed to have the same response options as the proxy report version [13]. In 8 of the 11 items, the 4 response categories were answered with substantial or moderate agreement. However, the level of agreement on difficulties with self-care, concentrating on things the child enjoys, and having changes to the routine were only fair. Similar problems with the item on difficulties with self-care were reported in an inter-rater reliability study between parents and teachers, which found poor agreement [35]. This would suggest that these items may have different meanings at different times of survey completion, and may need to be interpreted with caution [21]. 

Upon inspection of the item concerning “self-care”, one of the problems may be that the examples of self-care were presented in the item itself. The examples consisted of two different types of behavior, namely, eating or dressing up. The functions in relation to eating are vast. These can include fine motor coordination, such as the ability to use cutlery, means of swallowing, as well as desire to eat food. According to the ICF-child and youth version, there are five different codes related to just eating [10]. The other example of self-care—dressing up—may consist of differing functions. These may include gross motor coordination, such as putting arms through cloths, fine motor coordination to do up buttons or pull up zips, as well as other functions such as selecting clothes. Again, when mapped against the ICF, various different body functions as well as contextual factors are involved with this task of “self-care” [36]. Therefore, it may not be surprising that this item had low levels of reliability. It may be worthwhile to use only one concrete example of self-care that exemplifies child behavior. For the purpose of international comparability, modifications to the scale need to be explicitly stated when reporting the prevalence of children with self-care difficulties [13,14]. 

Another item with fair agreement levels was the item concentrating on things the individual enjoys. Children’s enjoyment of activities may change from one moment to another [37]. The instrument could easily be misinterpreted when there is a lack of consistency in behaviors being reported [31]. Naturally, the item was designed for reporting by the parents, and it is assumed the parents would know what the child enjoys doing [12]. However, this notion has been challenged as reported by Mactaggart and colleagues [15], who reported adults over-reported the functional difficulties from the child perception of difficulties. This could be because social interactions increase with peers and decline with family during adolescence [38]. More critical considerations are needed for this item when using both self-reports and when using a proxy among adolescents.

The item on concentrating was created as an extension of the Washington Group short set of six items, whereby one of item was related to “difficulties in remembering and concentration” [39]. The separation of items were not featured in an earlier draft reliability study (collected in 2015) of the CFM in a special education setting [35], indicating the possibility of a low level of evidence for the item. One observation of the items, as a whole, is that there are more child-related domains in the CFM and it is assumed they are uniform across the ages of 5–18 years old [15]. Although the CFM is divided into early childhood—between 2–4 years, there are no different question sets between pre- and post-puberty, or pre- and early adolescence. This may be regarded as a weakness of the CFM, and differences in either adult or child perceptions during adolescence may need to be encouraged for another separate package. An example of how survey items change is the Harter’s self-perception scale [40]. Harter’s scale was originally tested for 7–9-year-olds, and was later adapted for adolescents [41]. Perhaps such a convention is required for the self-report version of the CFM.

### 4.2. Reliability as Two Types of Cut-Offs

In reporting groups of children with disabilities, there are variable cut-off options. In our study, we carried out a test-retest on two cut-off values. The items on seeing, hearing, remembering things, controlling behavior, and getting friends could be interpreted to have had large agreement when the first level cut-off (at least some difficulties) was administered. These results partly contrast with the earlier evidence from the draft reliability tests of the CFM, whereby the levels agreements between parents and teachers were poor when reporting children who have at least some difficulty in getting friends [35]. Further evidence is needed that can be explained through triangulating the data from three main sources, the pupil, parents, and teacher before conclusions about these differing items can be made.

The first level cut-off value of at least some difficulties gives an indication of the number of adolescents who perceive any type of difficulties in performing the function. Aggregating this information may yield high prevalence of disability and may serve a purpose for providing indicators of trends over time. The second cut-off has been used as an indicator of disability prevalence in national-based studies [42]. In the case of hearing difficulties, there were not enough study participants to give a reliability statistic. There was large agreement in reporting behavior control difficulties, and moderate agreement for seeing, walking, and speaking difficulties. The cut-off points for disability prevalence in the functions of seeing, walking, speaking, and controlling own behaviors may be used among young adolescents in the special school environment.

This research offers a new insight into the way that children may self-report their own functional difficulties as an indicator for disabilities. The development of the work has been a long process, from the point of view of population statistics [11], to transfer the context for children [13], before it was converted to self-report for adolescents [43]. Through these steps, it would be possible to create data pooling for future big data sets. This could be a cost-effective answer to the problem where typically adult responses are a burden and group sizes of self-report are insufficiently large enough to make statistical comparisons and other analyses. For example, in the Finnish national monitoring study from over 6000 children and adolescents on physical activity behaviors, there were not enough cases to report difficulties in walking after stratifying by age and gender [44]. Children with walking difficulties are in an important group to report on as they have reportedly been considered to have the lowest levels of physical activity and could require adapted physical activities [29]. Although the results from our study may suggest that caution is required when interpreting some of the items, researchers and policy makers who use these items may need to consider which variables can actually be used to describe the prevalence of disabilities [45]. Another consideration would be to examine the grouping of individual items as a valid way to interpret data from the CFM, as illustrated by Zia and colleagues after they reported two factors from proxy reporting of the items [17]. Such considerations require further testing from various sources of information to advance the knowledge from data that have been disaggregated by disability.

### 4.3. Limitations

The sample was limited to children only in Finnish-speaking special schools in a region of Finland, and to those who were able to complete an online questionnaire. Different concepts of functional difficulties may exist in other environments, although there is the assumption that completion in the general school setting would yield improved reliability scores due less support needs. The study was on the intra-rater stability of the items, and reflects the perception of the child’s functional abilities, rather than corroborating with other data from other sources to validate the actual abilities, thus revealing discrepancies of the instrument and not to be considered as incidents of disabilities. It may be necessary to examine the construct and face validity of the items when interpreting the findings in future studies that adopt the child functioning module.

## 5. Conclusions

Self-reporting of functional difficulties is a subtle way of measuring childhood disabilities. There were large and moderate agreements when a cut off value of at least a lot of difficulties was used in four of the variables: seeing, walking, speaking, and controlling own behavior difficulties. Most of the self-report version of the Child Functioning Module can be used for adolescents, even if they have special or intensified support needs. The stability of two items (self-care and concentration) were poor and these items should be carefully considered in other studies. Further validation is required to support these findings.

## Figures and Tables

**Table 1 ijerph-17-06958-t001:** Items of the self-report Child Functioning Module in plain and easy-to-read English.

	Normative Language	Easy to Read Version
	Compared to children of the same age, do you have difficulties in…	Compare yourself to other teenagers of the same age. Which things are easy or difficult for you?
1	Seeing (even if you have to wear glasses or contact lens)?	Do you have difficulties in seeing (even if you have to wear glasses or contact lens)?
2	Hearing (even if you have a hearing aid)?	Do you have difficulties in hearing (even if you have a hearing aid)?
3	Walking 100m, for example a length of a football pitch (even if you use assistance)?	Do you have difficulties in walking 100m, for example the length of a football pitch (even if you use assistance)?
4	Self-care, for example eating or dressing up?	Do you have difficulties in self-care, for example eating or dressing up?
5	Being understood when speaking (outside of the home)?	Do you have difficulties in being understood when speaking to people outside of your home?
6	Learning things?	Do you have difficulties in learning things?
7	Remembering things?	Do you have difficulties in remembering things?
8	Concentrating on things you enjoy?	Do you have difficulties in concentrating on things you enjoy?
9	Making changes to your own routine?	Do you have difficulties in making changes to your own routine?
10	Controlling your own behaviors?	Do you have difficulties in controlling you own behaviors?
11	Getting friends?	Do you have difficulties in getting friends?

**Table 2 ijerph-17-06958-t002:** Description of the breakdown of participants in the study.

Variable	M-er	M	L-er	L	Total
Total	6	54	5	9	74
Boys	5	32	3	7	47
Girls	1	22	2	2	27
Disabilities (% some or more)	4 (67%)	31 (56%)	5 (100%)	5 (55%)	57 (65%)
Age in years (SD)	14.8 (1.2)	12.3 (1.5)	16.1 (0.5)	15.6 (0.6)	13.2 (1.8)

**Table 3 ijerph-17-06958-t003:** Distribution of reporting difficulties from retest survey and prevalence using the cut-points based at some difficulties and a lot of difficulties.

Difficulties in…	None	Some	A Lot	Cannot do	Missing	Some + %	A Lot + %
Seeing	66	6	0	2	0	10.8	2.7
Hearing	66	7	0	1	0	10.8	1.4
Walking	68	4	0	2	0	8.1	2.7
Self-Care	65	7	1	1	0	12.2	2.7
Speaking	63	8	2	1	0	14.9	4.1
Learning	50	19	3	2	0	32.4	6.8
Remembering	51	20	2	1	0	31.1	4.1
Concentration	58	14	0	1	1	20.5	1.4
Routine	56	15	1	1	1	23.3	2.7
Behavior	56	14	1	2	1	23.3	4.1
Friends	58	10	2	3	1	20.5	6.8

**Table 4 ijerph-17-06958-t004:** Intra-rater reliability results of self-reported child functioning module.

Function	ICC	LCI	UCI	Landis and Koch	Kappa 1	*p*	Cohen	Kappa 2	*p*	Cohen
Seeing	0.753	0.608	0.845	Substantial	0.801	<0.001	Large	0.486	<0.001	moderate
Hearing	0.63	0.411	0.768	Substantial	0.457	<0.001	Moderate	n/a		
Walking	0.645	0.449	0.782	Substantial	0.509	<0.001	Large	0.486	<0.001	moderate
Self-Care	0.345	−0.027	0.585	Fair	0.251	0.018	Small	0		poor
Speaking	0.76	0.619	0.849	Substantial	0.472	<0.001	Moderate	0.490	<0.001	moderate
Learn	0.59	0.35	0.742	Moderate	0.410	<0.001	Moderate	0.210	0.061	small
Remembering	0.625	0.405	0.764	Substantial	0.563	<0.001	Large	−0.210	0.836	small
Concentration	0.204	−0.274	0.502	Fair	0.28	0.016	Small	−0.019	0.866	poor
Routine	0.277	−0.145	0.544	Fair	0.341	0.003	Moderate	−0.034	0.767	poor
Behavior	0.732	0.573	0.832	Substantial	0.548	<0.001	Large	0.652	<0.001	large
Friends	0.57	0.314	0.73	Moderate	0.531	<0.001	Large	0.256	0.015	small

ICC—Intraclass correlation coefficient, LCI—lower confidence interval, UCI—upper confidence interval, Kappa 1—Kappa based on cut-point of “at least some difficulties”, Kappa 2—Kappa based on cut-point of “at least a lot of difficulties”, n/a—not available

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
