# Peer review of "Intra-Rater Test-Retest Reliability of a Modified Child Functioning Module, Self-Report Version"

_ijerph, 2020, doi:10.3390/ijerph17196958_

Round 1

Reviewer 1 Report

The study does well in terms of contributing to the determination of disability prevalence as an important area for population statistics globally, especially among young adolescents. Explaining the context of the Finnish education system, the researchers consciously aim to eliminate/reduce adults’ interpretation of children’s own perceptions and giving young disabled adolescents themselves a say, enabling them to self-report this measure. This is an overwhelmingly positive innovation. The research design of allowing a 2-week interval before re-testing seems appropriate, though the issue of children with hearing impairments not participating in the second round (line 224)  would benefit from explanation, in the interests of inclusion. The conclusions drawn and the cautions expressed are appropriate to the study.

This research offers new insight into the way children may self-report their own functional difficulties as an indicator for disabilities (lines 304-5).  In particular, it is highly appropriate that the study includes the conclusion that there is a need to include self-reported disabilities in national health surveys (lines 236-7).

Within this broad positive context, there are however a number of areas which would require further discussion.

  1. Control group

The paper does not refer to a control group. This would provide a useful check with regard to some of the questions asked, such as ‘Do you have difficulties in concentrating on things you enjoy?’.

  1. Consent

The study does not sufficiently outline how it addressed ethical issues in research with children, based on a children’s rights approach which is now to be expected in all research with children. For example, were children asked at the start of each session if they felt okay about participating, that nothing they said would be reported to their teachers or parents, etc.

  1. Medical model versus social model

The study is based on the Washington Group on Disability Statistics, one source of reporting disabilities through functional difficulties. It consequently focuses on ‘child functioning module (CFM) which it uses in suitably modified formats, allowing children to self-report.

This approach is firmly embedded in a medical perspective. Although this has for the past two decades or so been critiqued, mainly by disabled people themselves and their representative organisations, the paper does not engage with this at any point. The medical model links a disability diagnosis to an individual's physical body, reflecting the view that ‘the disability is in you and it is your problem’. By contrast, the social model of disability reflects the problem lying in the interaction between the individual and society. It regards disability issues stemming from someone with a disability trying to function in an inaccessible society and being able to participate fully in day to day life.

It would enhance the study if the researchers were to engage with this issue, even if only to explain why the medical model was being applied, and demonstrating awareness of its limitations.

  1. Qualitative vs quantitative survey

The study states that ‘schools form a good place to recruit children for important health checks as well as carry out health surveys ‘ (lines 47-48). The setting of a supervised classroom seems convenient for the researchers, but they do not discuss or address the potential impact of this setting, with their teachers and some parents present, on the participants.

In addition, only four specific answers were provided which children could choose from.

A qualitative approach would undoubtedly unearth more nuanced and therefore more valuable responses. For example, Connors (2003) stated that: “We asked the children to tell us about ‘a typical day’ at school and at the weekend, relationships with family and friends, their local neighbourhood, experiences at school, pastimes and interests, use of services and future aspirations . . .  open-ended questions were enough to launch some children on a blow-by-blow account . . .”. This study found that “children experienced disability in four ways, in terms of impairment, difference, other people's reactions, and material barriers.” Connors, C. and Stalker, K (2003) The views and experiences of disabled children and their siblings: a positive outlook. (London, Jessica Kingsley),

The responses are therefore not surprising, as disabled children tend to see impairment in medical terms because many have a significant amount of contact with health services. But as Cavet (1998) points out, at this age, social activities like leisure and friendship happen either in young people's homes or venues like sporting facilities or shopping centres, neither of which may be accessible to some disabled adolescents and teenagers. Cavet J. (1998) Leisure and friendship, in: C. Robinson and K. Stalker (Eds.) Growing up with Disability (London, Jessica Kingsley)

Applying the medical model and a quantitative approach demonstrates the weaknesses of the methodology, as significant aspects of potential self-reporting are necessarily missed which is regrettable.

Author Response

Reviewer 1:

The study does well in terms of contributing to the determination of disability prevalence as an important area for population statistics globally, especially among young adolescents. Explaining the context of the Finnish education system, the researchers consciously aim to eliminate/reduce adults’ interpretation of children’s own perceptions and giving young disabled adolescents themselves a say, enabling them to self-report this measure. This is an overwhelmingly positive innovation. The research design of allowing a 2-week interval before re-testing seems appropriate, though the issue of children with hearing impairments not participating in the second round (line 224)  would benefit from explanation, in the interests of inclusion. The conclusions drawn and the cautions expressed are appropriate to the study.

This research offers new insight into the way children may self-report their own functional difficulties as an indicator for disabilities (lines 304-5).  In particular, it is highly appropriate that the study includes the conclusion that there is a need to include self-reported disabilities in national health surveys (lines 236-7).

Within this broad positive context, there are however a number of areas which would require further discussion.

RESPONSE: Thank you for the positive comments provided for this review. We have addressed your specific comments below

  1. Control group

The paper does not refer to a control group. This would provide a useful check with regard to some of the questions asked, such as ‘Do you have difficulties in concentrating on things you enjoy?’.

RESPONSE: There was no control group to measure from. The items were from the Washington Group Child Functioning module and were transformed from parent proxy reporting to self-report. As such, this is one of the reasons we choose to test the psychometric properties of these items through an intra-rater reliability test so we can check how well these questions can be used for monitoring purposes. 

  1. Consent

The study does not sufficiently outline how it addressed ethical issues in research with children, based on a children’s rights approach which is now to be expected in all research with children. For example, were children asked at the start of each session if they felt okay about participating, that nothing they said would be reported to their teachers or parents, etc.

RESPONSE: We agree with the reviewer that ethical considerations are important in this setting than in a general school setting. We described and followed through the study protocol that was approved by our institutional ethical review committee (L84). As part of the protocol, we had to obtain consent to take part, and we informed the students that they do so voluntarily (L103). We were aware of potential bias in reporting in the presence of teachers, hence in our protocol we explained to teachers what their role was during data collection. We also asked the teachers to complete their own survey at the same time to give the impression to the students that the teacher was also busy.  These were outlined on lines 103-4.

  1. Medical model versus social model

The study is based on the Washington Group on Disability Statistics, one source of reporting disabilities through functional difficulties. It consequently focuses on ‘child functioning module (CFM) which it uses in suitably modified formats, allowing children to self-report.

This approach is firmly embedded in a medical perspective. Although this has for the past two decades or so been critiqued, mainly by disabled people themselves and their representative organisations, the paper does not engage with this at any point. The medical model links a disability diagnosis to an individual's physical body, reflecting the view that ‘the disability is in you and it is your problem’. By contrast, the social model of disability reflects the problem lying in the interaction between the individual and society. It regards disability issues stemming from someone with a disability trying to function in an inaccessible society and being able to participate fully in day to day life.

It would enhance the study if the researchers were to engage with this issue, even if only to explain why the medical model was being applied, and demonstrating awareness of its limitations.

RESPONSE: Although the items are on functions, it is our understanding the CFM items are based on the ICF [1], provided by the WHO [2]. Under the UN definition of disabilities, “Persons with disabilities include those who have long-term physical, mental, intellectual or sensory impairments which in interaction with various barriers may hinder their full and effective participation in society on an equal basis with others” [3], which too was based on the ICF.  Based on this understanding of the Washington Group (see paper – ICF[4]), we would disagree with the reviewer that it is based on the medical model. The CFM is sensitive to, and designed to avoid stigma related to the notion of “disabled or not”, and forces the individual to focus on the functions. We would encourage a commentary about this in response to the suggestions by the reviewer, but we also feel that within this paper, going into the details of disability may detract from the main focus of the paper, of which, is on the psychometric properties of these items for use in national surveys.

Qualitative vs quantitative survey

The study states that ‘schools form a good place to recruit children for important health checks as well as carry out health surveys ‘ (lines 47-48). The setting of a supervised classroom seems convenient for the researchers, but they do not discuss or address the potential impact of this setting, with their teachers and some parents present, on the participants.

In addition, only four specific answers were provided which children could choose from.

A qualitative approach would undoubtedly unearth more nuanced and therefore more valuable responses. For example, Connors (2003) stated that: “We asked the children to tell us about ‘a typical day’ at school and at the weekend, relationships with family and friends, their local neighbourhood, experiences at school, pastimes and interests, use of services and future aspirations . . .  open-ended questions were enough to launch some children on a blow-by-blow account . . .”. This study found that “children experienced disability in four ways, in terms of impairment, difference, other people's reactions, and material barriers.” Connors, C. and Stalker, K (2003) The views and experiences of disabled children and their siblings: a positive outlook. (London, Jessica Kingsley),

The responses are therefore not surprising, as disabled children tend to see impairment in medical terms because many have a significant amount of contact with health services. But as Cavet (1998) points out, at this age, social activities like leisure and friendship happen either in young people's homes or venues like sporting facilities or shopping centres, neither of which may be accessible to some disabled adolescents and teenagers. Cavet J. (1998) Leisure and friendship, in: C. Robinson and K. Stalker (Eds.) Growing up with Disability (London, Jessica Kingsley)

Applying the medical model and a quantitative approach demonstrates the weaknesses of the methodology, as significant aspects of potential self-reporting are necessarily missed which is regrettable.

RESPONSE: We acknowledge the reviewer’s perspective and the child’s voice is indeed a valuable source of information. However, the debate that the reviewer presents most likely detracts away from the main focus of the paper, which is the psychometric properties (through intra-rater reliability) of the CFM for the purpose of national monitoring surveys. The purpose of survey instruments is not to find out the in-depth experiences of the respondents, as that would require a different study design, which is known to also have its limitations of sampling bias, and representativeness from willing parties, trustworthiness of interpretation [5]. We position ourselves in a positivism frame for using quantitative data with the potential to gather data through anonymous surveys, and thus can create comparable data between population groups. The first note about the influence of the presence of the teacher when carrying out the survey was considered and hence we created a distraction strategy whereby the teachers was seen to be doing their own survey at the same time. This gave the opportunity for a sense of anonymity and reduces the bias for responses, assuming that students in special educational needs settings would report biases. We doubt there is a perfect solution to the perspectives, and believe the methodology depends on the purposes of the studies.

In the case of developing instruments such as the WG, which, we reiterate, was not based on the medical model, but rather the biopsychosocial model (see WHO ICF for details), it is one step towards the possibility of data disaggregation for national surveys, including the child’s voice in a comparable and anonymous way, as well as improving policy decisions that are universal and supported by evidence. As mentioned from the previous comment, we feel that the comments from the reviewer do warrant commentary on, but perhaps out of the scope of this paper. We would welcome discussing this further should the editor allow, in a separate commentary or response, and the reviewer feels inclined to take up this invitation. Therefore, at this juncture, I hope the reviewer’s comments and concerns are addressed in keeping the manuscript with a main focus to the evidence presented from the study.

References

  1. Loeb, M.; Mont, D.; Cappa, C.; De Palma, E.; Madans, J.; Crialesi, R. The Development and Testing of a Module on Child Functioning for Identifying Children with Disabilities on Surveys. I: Background. Disabil Health J 2018, 11, 495-501.
  2. WHO. International Classification of Functioning, Disability and Health (ICF).; World Health Organization: Geneva, Switzerland, 2001.
  3. United Nations. Convention on the Rights of Persons with Disabilities and Optional Protocol. 2006, A/RES/61/106.
  4. Kostanjsek, N. Use of the International Classification of Functioning, Disability and Health (ICF) as a Conceptual Framework and Common Language for Disability Statistics and Health Information Systems. BMC Public Health 2011, 11, S3.
  5. Tenenbaum, G.; Gershgoren, L.; Schinke, R.J. Non-Numerical Data as Data: A Positivistic Perspective. Qualitative Research in Sport, Exercise and Health 2011, 3, 349-361.

Reviewer 2 Report

Thank you for send me this review. It's a great job and very important to know more about disability children and their functionality.

The article complies with the norms of a good summary. In addition, it is well structured. The introduction is brief and places in the problem that is approached in the current world of disability, the problems that exist when evaluating certain fields. They have followed the steps established by the WHO for this type of study, in addition to adapting the questionnaires in "easy reading" for disabled children. The analysis is adequate as are the conclusions.

Author Response

Thank you for the feedback for the paper. We have added extra information in the introduction as suggested by the other reviewers. 

Reviewer 3 Report

Thank you very much for having the opportunity to review this manuscript.

The article presented for review: Intra-rater test-retest reliability of self-reported child functioning module
concerns the important methodological and social issue of the reliability of tools for measuring reported difficulties in functioning of people with disabilities and intended for them. The article has been correctly formatted and divided into parts as required by the journal. It has been correctly written and has appropriately selected literature. The authors correctly presented the research topic, which was presented to the reader in an accessible form. The article is an important and innovative, on the European basis, scientific analysis of the study of the psychometric properties of tests for measuring difficulties in functioning of disabled people. It can be applied in practical outcomes in the work of educators, psychologists and doctors. Although I consider this topic to be an extremely important one, the diligence of the study before publication requires some, in my opinion significant corrections that should be introduced to the text in the following scope and parts of the article:

Title
• it is worth taking into account the Finnish research context in the title, so that for the reader the clarity of the intra-rater test-retest reliability of self-reported child functioning module applies to the Finnish research sample

• the title in the current formula containing the word child is, in my opinion, misleading because the content of the article and the research concerned students in the developmental period of early adolescence. Early adolescence is a different / next stage of human development than childhood. In the opinion of the reviewer, it is not appropriate to use the word "child" when the analyzes concern young adolescents

• in the title, the reader should be introduced to the general context of the content of the article and at the same time emphasize its strength - intra-rater test-retest reliability study on a modified set of items for self-reporting functional difficulties in young student with disability

Abstract

• line 18-19 is (n = 74; boys = 64%; age m = 13.7; SD = 1.8) should be (N = 74; boys = 64%; age M = 13.7; SD = 1.8)

Keywords
• I propose to reduce the number of keywords into 3-5 and choose those that match the content of the article the best

Introduction
• it is worth extending the literature analysis to include the context of comparable research (test-retest) conducted in other countries using The Washington Group on Disability Statistics
• a short description of the developmental period of early adolescence should be made, with particular emphasis on the development of disabled adolescents and reference to the areas of functioning which are the subject of research in the article

• it is worth describing the current gap in the literature that justifies the need for conducting this study

Materials and Methods

  • line 83-84 is not clearly written to what the size of the research sample indicated by the test refers to - should there be 67 schools selected for the research or of the respondents? - please specify this provision
  • line 86 in the opinion of the reviewer (NL) is not necessary
  • line 106-110 - please specify which questionnaire was completed by a student who was 14 years old (if students of this age participated in the study). This is not specified in the current record. The authors state that the age of the respondents was between 11 and 15 years (line 90) and that the shortened version of the questionnaire was completed by people aged 11-13 and the longer - 15 years.
  • line 155, in the opinion of the reviewer, the records: (KN, PR, NL, PA) are not necessary
  • line 154-155 on what basis the authors of the article decided that particular items translated into Finnish correspond to the original (English language). How was the entire process of adapting items from English to Finnish? The basis for a reliable examination of the credibility of a test from a different culture is the process of its adaptation, including translation (for exemple see Adapting Educational and Psychological Tests for Cross-cultural Assessment, eds. R.K. Hambleton, P.F. Merenda, CD. Spielberger, Lawrence Erlbaum, Hillsdale 2005). There is no record in the text of the article about the course of the adaptation procedure for individual items
  • the novelty of this article are self-reporting functional difficulties studies conducted among young people, however, the authors do not provide the basic characteristics of the respondents (age, sex,% of people with disabilities). Please complete the article in the Materials and Methods section with a separate subsection characterizing the group of participants. The description of the study group contained in Table 2 is insufficient
  • the entry on line 189-191 is unnecessary
  • it is worth providing (if it was measured) the time in which people answered the questions or write down that this variable was not taken into account in the analyzes

Limitations
• it is worth adding to the listed test limitations those related to the use of a given method to assess the reliability of the test (test-retest)

Discussion
• I propose a greater reference than before to the research that was cited in the theoretical part of the article, but are not discussed in the context of own results

• a comparative discussion of the obtained results with similar studies (if any) from other countries should be conducted

Author Response

Thank you very much for having the opportunity to review this manuscript.

The article presented for review: Intra-rater test-retest reliability of self-reported child functioning module
concerns the important methodological and social issue of the reliability of tools for measuring reported difficulties in functioning of people with disabilities and intended for them. The article has been correctly formatted and divided into parts as required by the journal. It has been correctly written and has appropriately selected literature. The authors correctly presented the research topic, which was presented to the reader in an accessible form. The article is an important and innovative, on the European basis, scientific analysis of the study of the psychometric properties of tests for measuring difficulties in functioning of disabled people. It can be applied in practical outcomes in the work of educators, psychologists and doctors. Although I consider this topic to be an extremely important one, the diligence of the study before publication requires some, in my opinion significant corrections that should be introduced to the text in the following scope and parts of the article:

RESPONSE: Thank you for your positive comments regarding the manuscript. We are responding to each suggestion for corrections below.

Title
• it is worth taking into account the Finnish research context in the title, so that for the reader the clarity of the intra-rater test-retest reliability of self-reported child functioning module applies to the Finnish research sample

RESPONSE: This is a consideration that we looked into, and we trust that the reviewer is willing to accept that we do not change the title but have ensured there is emphasis in the abstract that the data is from Finland in the Abstract. After looking at the top 10 papers from google scholar, “https://scholar.google.com/scholar?hl=en&as_sdt=0%2C5&as_ylo=2017&q=Intra-rater+test-retest+reliability+survey&btnG=, none had the location of data in the title. We do understand that context is important when considering these results, hence we have placed this information in the abstract.

  • the title in the current formula containing the word child is, in my opinion, misleading because the content of the article and the research concerned students in the developmental period of early adolescence. Early adolescence is a different / next stage of human development than childhood. In the opinion of the reviewer, it is not appropriate to use the word "child" when the analyzes concern young adolescents

RESPONSE: The reviewer has a very good point. We have used terminology in the manuscript of ‘young adolescents’, as suggested by Sawyer and colleagues (2017). However, the term ‘child’ comes from the instrument that we are conducting the test on. The Child Functioning Module has been capitalised in the title to reflect it is a recognised instrument. We acknowledge the limitation of the age range in the discussion section, on L292-294, highlighting further the need for scientific work like ours to report the evidence on the psychometric properties of the instrument.

  • in the title, the reader should be introduced to the general context of the content of the article and at the same time emphasize its strength - intra-rater test-retest reliability study on a modified set of items for self-reporting functional difficulties in young student with disability

RESPONSE: We like to thank the reviewer’s suggestion for the title, and adopted the ‘modified’ part to reflect the title. Although we do not use the ‘young student with disability’ as we believe it is important to emphasis this is developmental work on the CFM.

Abstract

  • line 18-19 is (n = 74; boys = 64%; age m = 13.7; SD = 1.8) should be (N = 74; boys = 64%; age M = 13.7; SD = 1.8)

RESPONSE: Thank you for pointing this out. The capitalised N and M have been included, as well as spaces around the equals sign.

Keywords
• I propose to reduce the number of keywords into 3-5 and choose those that match the content of the article the best

RESPONSE: We would like the possibility to keep all these keywords that do not appear in the title. Given the journal template states, “(List three to ten pertinent keywords specific to the article; yet reasonably common within the subject discipline.)”, we hope the reviewer does not see this being a problem.

Introduction
• it is worth extending the literature analysis to include the context of comparable research (test-retest) conducted in other countries using The Washington Group on Disability Statistics

RESPONSE: We have included some information about the prior work on the testing of the WG in the introduction on L61-64, by stating;

“Additional functions that are crucial in the development of children under the age of 18y that are distinctly different from adulthood included psychosocial development items in learning, behavioural control as well as social interactions to be featured in the CFM”.

• a short description of the developmental period of early adolescence should be made, with particular emphasis on the development of disabled adolescents and reference to the areas of functioning which are the subject of research in the article

RESPONSE: We have included some literature about the extension of the WG items to other functions that are particular for children to produce the child functioning module around L61-4 by stating;

“Additional functions that are crucial in the development of children under the age of 18y that are distinctly different from adulthood included psychosocial development items in learning, behavioural control as well as social interactions to be featured in the CFM”..

  • it is worth describing the current gap in the literature that justifies the need for conducting this study

RESPONSE: The description of the current gap in the literature is stated in the aim of this study on L79;

“and the lack of psychometric properties available from the self-report version of disabilities”:

Materials and Methods

  • line 83-84 is not clearly written to what the size of the research sample indicated by the test refers to - should there be 67 schools selected for the research or of the respondents? - please specify this provision

RESPONSE: We have made it clearer than the sample of 67 students was required.

  • line 86 in the opinion of the reviewer (NL) is not necessary

RESPONSE: Point taken, and we suggest this is decided by the editor for whether to leave it in or take it out.

  • line 106-110 - please specify which questionnaire was completed by a student who was 14 years old (if students of this age participated in the study). This is not specified in the current record. The authors state that the age of the respondents was between 11 and 15 years (line 90) and that the shortened version of the questionnaire was completed by people aged 11-13 and the longer - 15 years.

RESPONSE: We wrote this paper to report the age similarities to other international and national surveys carried out the by the researchers (L95), although in practice administration of surveys are carried out by school grade. Samples are then checked for mean ages and then categories into the 11y, 13y, and 15y age groups for comparisons between country studies. The age groups for special education are slightly different in that they do not necessarily correspond directly to the grade as some students need a bit more time before advancing to the next grade. Hence we have used the term ‘equivalent to’ (L116-9) in the description, and carry on with a short explanation of the complexity with physical age, compared with developmental age.

  • line 155, in the opinion of the reviewer, the records: (KN, PR, NL, PA) are not necessary

RESPONSE: Point taken, and we suggest this is decided by the editor for whether to leave it in or take it out.

  • line 154-155 on what basis the authors of the article decided that particular items translated into Finnish correspond to the original (English language). How was the entire process of adapting items from English to Finnish? The basis for a reliable examination of the credibility of a test from a different culture is the process of its adaptation, including translation (for exemple see Adapting Educational and Psychological Tests for Cross-cultural Assessment, eds. R.K. Hambleton, P.F. Merenda, CD. Spielberger, Lawrence Erlbaum, Hillsdale 2005). There is no record in the text of the article about the course of the adaptation procedure for individual items

RESPONSE: Thank you for this observation and the suggested reference by Hambleton and colleagues, 2005. We have followed the work by Bergendorff, 2009 for ensuring translations were accurate. In addition, the translations built from previous work (which we have acknowledged in the text) guided by the documentation from the Washington Group.

  • the novelty of this article are self-reporting functional difficulties studies conducted among young people, however, the authors do not provide the basic characteristics of the respondents (age, sex,% of people with disabilities). Please complete the article in the Materials and Methods section with a separate subsection characterizing the group of participants. The description of the study group contained in Table 2 is insufficient

RESPONSE: We have followed the guidance from the STROBE checklist for cross-sectional studies in reporting population characteristics. Under methods – participants, we report the “method of selection of participant”  and in the results section, we “report numbers of individuals at each stage of study”, and “give characteristics of study participants”. Therefore, unless requested by the editors, we would like to keep the reporting in their place.

The second issue brought up by the reviewer is providing details of the study characteristics. The trouble with reporting the disability is that this is the dependent variable for the study. This is the variable that we are testing, and therefore we are unsure if the instrument is sound enough to report them if we are reporting the uncertainty of what is being based upon. As mentioned in the limitations of the study, we did not corroborate with other data sources. We have however reported the numbers of the students who completed different surveys and that was to provide some insight into the severity of disabilities and age by the respondents. These were broken down by counts. As these are descriptive in nature of typical disaggregation variables, no statistical tests were conducted, hence Table 2 was not modified. We trust this explanation is suitable for the reviewer’s consideration.

  • the entry on line 189-191 is unnecessary

RESPONSE: Thank you for the observation, it has now been removed (lines underneath this may have then changed).

  • it is worth providing (if it was measured) the time in which people answered the questions or write down that this variable was not taken into account in the analyses

RESPONSE: Although time for completion may have an influence on the quality of response, it was not possible to measure. This was unfortunate and the reason for this was that it was not a single survey instrument, rather it was a multi-faceted questionnaire on health and physical activity behaviours. We included a statement that this was not possible to analyse in L186-7 and wrote;

“Time for completion was not analysed as this was not available due to there being other questions in the overall survey”.

Limitations
• it is worth adding to the listed test limitations those related to the use of a given method to assess the reliability of the test (test-retest)

RESPONSE: We have included extra text on lines 334-5 to state further limitation from the survey and study;

“thus revealing discrepancies of the instrument and not to be considered as incidents of disabilities”.

Discussion
• I propose a greater reference than before to the research that was cited in the theoretical part of the article, but are not discussed in the context of own results

RESPONSE: We agree with this point made by the reviewer and added extra reflexive discussion based on the work by Marker, Steele and Noser on line 248-249, stating;

“Young adolescents is an important time for reflecting on personal growth and these changes must be monitored”.

  • a comparative discussion of the obtained results with similar studies (if any) from other countries should be conducted

RESPONSE: We would like to inform the reviewer that we have conducted searches to see if this has been carried out in other contexts (countries, age groups, schools) and did not find any report to compare against. We have, reported similar studies based on other versions of the instrument, such as the parent and teacher version, how those results faired and were reported around L259-260, L286-288. We have a statement, to suggest that this is the first study of its kind (on line 244), and thus limiting its ability to discuss with other available research.

Round 2

Reviewer 1 Report

I would like to thank the authors for their detailed responses to the areas I had identified as problematic. Nevertheless I feel that there is not sufficient move to address these concerns in the paper. I think it could be improved by explicitly mentioning the areas identified in the paper itself, and providing the authors' responses. In this way the readers would be able to judge for themselves whether the responses are adequate. 

Author Response

Dear reviewer, 

Thank you once again for the time to review the responses and the revised paper. It is currently unclear what the reviewer would like in this review round. We are not sure how the readers can be judges of adequacy of responses as there were no explicit requests this time around. 

As mentioned in the first round reviews, the authors feel that some of the requests were beyond the scope of the paper, and would welcome a separate commentary, in another paper, should the editor permit. The areas that the reviewer brings up are relevant debates to advance the discourse in disability research, yet we would like to politely remind the reviewer, the purpose of the paper is to report the results of the test-retest of the instrument. 

The timing of the request is timely after the recent publication by Zia and colleagues (Zia, N., Loeb, M., Kajungu, D. et al. Adaptation and validation of UNICEF/Washington group child functioning module at the Iganga-Mayuge health and demographic surveillance site in Uganda. BMC Public Health 20, 1334 (2020). https://doi.org/10.1186/s12889-020-09455-1). We have incorporated where we felt it it fitted in the paper to keep the paper at the forefront of the research. 

Reviewer 3 Report

Thank you for all the explanations of the authors of the article regarding the areas indicated by the reviewer in the content of the article that required correction.
These explanations are sufficient for me.
I recommend an article for publication in this journal.
Yours sincerely

Author Response

The authors would like to thank the reviewer for your recommendation.